# Fluid migrations and volcanic earthquakes from depolarized ambient noise

S. Petrosino[1] & L. De Siena [2,3 ✉]

Ambient noise polarizes inside fault zones, yet the spatial and temporal resolution of polarized noise on gas-bearing fluids migrating through stressed volcanic systems is unknown. Here we show that high polarization marks a transfer structure connecting the deforming centre of the caldera to open hydrothermal vents and extensional caldera-bounding faults during periods of low seismic release at Campi Flegrei caldera (Southern Italy). Fluids pressurize the Campi Flegrei hydrothermal system, migrate, and increase stress before earthquakes. The loss of polarization (depolarization) of the transfer and extensional structures maps pressurized fluids, detecting fluid migrations after seismic sequences. After recent intense seismicity (December 2019-April 2020), the transfer structure appears sealed while fluids stored in the east caldera have moved further east. Our findings show that depolarized noise has the potential to monitor fluid migrations and earthquakes at stressed volcanoes quasi-instantaneously and with minimum processing.

[1] Istituto Nazionale di Geofisica e Vulcanologia, Sezione di Napoli - Osservatorio Vesuviano, Napoli 80124, Italy. [2] Institute of Geosciences, Johannes Gutenberg University, Mainz 55128, Germany. [3] TeMaS - Terrestrial Magmatic Systems Research Area, Johannes Gutenberg University, Mainz 55128, Germany. ✉email: ldesiena@uni-mainz.de

We have learned how to use noise produced by humans[1], ocean swell, and atmosphere solid-Earth interactions[2] to illuminate the interior of magmatic[3,4] and hydrothermal systems[5,6]. Noise data from expanding seismic networks are analyzed with novel array[7] and interferometric[8,9] techniques, allowing detection of volcanic processes and forecasting hazards without having to wait for earthquakes[10,11]. Ambient noise can show complex polarization[12,13], i.e., preferential directions and planes of oscillations, especially when higher modes of surface waves and body waves mix with fundamental modes. Noise polarization provides information on oceanic processes[2] and can be related to stress and variations in stiffness anisotropy across faults[12,13]. However, polarization studies generally measure azimuths where polarization is highest, relating them to Earth's processes and structures[13]. The loss of polarization (depolarization) has never been employed to monitor deep fluid-induced dynamics. Once applied to stressed volcanic structures, depolarization could provide a new way to monitor volcanic activity and associated earthquakes.

Campi Flegrei (Southern Italy, Fig. 1a, small lower panel) is an inhabited volcanic caldera bordering Naples (the third most populous city of Italy) and the ideal location to highlight the potential of depolarized noise to monitor volcanoes. The caldera is a capped geothermal system[14–16], where hazardous pressurized fluids propagate from the primary deformation source[17] (Figs. 1–3, black dot) to fumaroles (S) at least since 1984[18,19]. Heating of the hydrothermal system, volcanic gas emissions at the surface, and seismic release result from consecutive episodes of unrest, promoting a long-term accumulation of lateral stress and expanding reservoirs[20,21]. Analogue modelling[22], seismic tomography[23], and extensive geological fieldwork[24–26] conclude that NW-SE-trending extensional and caldera-bounding faults bear most of the regional stress north and east of the caldera (Fig. 1a, white dotted line). Fieldwork, numerical modelling[22], and deformation inversions[27] also infer the existence of NE-SW-trending transfer structure, feeding volcanic activity connecting primary deformation source and degassing vents[28,29] (Fig. 1a, black dotted line). Geophysical imaging methods have never imaged this transfer structure.

Both extensional faults and transfer structure were likely crucial for developing volcanic unrests monitored during the last thirty-six years. On April 1st, 1984, an NW-directed injection of magmatic or supercritical fluids opened a low-velocity hydrothermal reservoir located in the centre of Campi Flegrei caldera[18] in the WSW-ENE direction[16]. The injection location was estimated from the source characteristics and spatial relation of repeated vertically-aligned earthquakes (black diamonds, Fig. 2a)[16], whose timing was compared with geochemical data and the results of fluid and heat flow modelling[19]. The injection location corresponds to the point of maximum coda attenuation in the caldera[29] (Fig. 2a). After thirty years (2011–13), the low-velocity reservoir had expanded from the injection point, becoming aseismic[6] (Fig. 2b). Expansion toward west and north continued until fluids had reached the western caldera-bounding faults, producing seismic swarms in 2012[30] (Fig. 2b, western black diamonds). However, no apparent lateral expansion was visible east and south of the injection point (black cross, Fig. 2a, b). Fluids stopped at a barrier delineated by high velocities and high stresses, as shown by combined seismic and InSAR interferometric analyses[31]. Here, InSAR[31], shear-wave-splitting anisotropy[32], gravity gradiometry[33], and strong seismic velocity contrasts[6,34] identify an SN anomaly that accumulates the highest lateral stress during the unrest, producing small-magnitude earthquakes[35] (Fig. 2b, c, white ellipse, the eastern sector of Solfatara).

This study measures and maps noise polarization attributes at Campi Flegrei using data recorded across years, months, and days between 2009 and 2020. We compare the maps with geological, geophysical, geodynamical, and volcanological information, separating periods of lower and higher seismic release. Our results show that polarized noise detects both the extensional faults and the transfer structure at the caldera during periods of low seismic release. The depolarization of the transfer structure marks both injections at the start of seismic unrest and lateral fluid migrations leading to earthquakes. The results detect structures and processes leading to hazard at Campi Flegrei caldera, offering a new technique to monitor fluid-derived processes across highly-stressed volcanoes in real time.

## Results
**Spatial polarization measurements.** The azimuth of the horizontal polarization vector derived from ambient noise and the resultant length of its distribution $(R)$[12,13,36,37] are used here for the first time as both imaging and diagnostic tools (Methods). During periods of low seismic release[38,39], they detect the hypothesized link between the extensional faults that bear regional stress north and east of the caldera (Fig. 1a, white dotted line) and a dynamic transfer structure[22,27] that crosses its deforming centre and vents

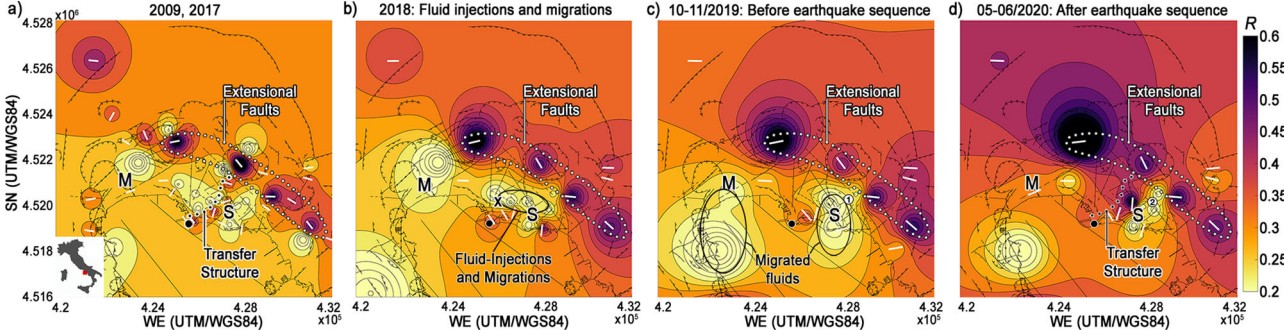

**Fig. 1 Maps of resultant lengths and azimuths from ambient noise at Campi Flegrei. a** The resultant length ($R$) is plotted with a squared interpolation from each station between 0.2 and 1 Hz during periods of low seismic release (2009, 2017). The continuous white segments show the corresponding azimuths (only for $R > 0.25$). The patterns are imposed over mapped fault strikes, fractures, and craters[24–26]. The Solfatara crater (S) and Monte Nuovo (M) are marked on the maps. The wide black dot is the stationary point of maximum vertical deformation for the last 36 years[27,31]. The dotted black line marks the transfer structure ($R > 0.31$). The dotted white line contours the portion of the NW-SE extensional faults that shows R > 0.5 and the same azimuths over a decade. **b** Same map obtained using noise recorded over six months in 2018. The black cross shows the centre of the high-attenuation anomaly in Fig. 2a. Part of the transfer structure depolarizes due to fluid injections and migrations (black ellipse). **c** Same map obtained using two months of noise recorded before the Md3.1 earthquake (December 6th, 2019, circled number 1) after fluids migrated to the east and west reservoirs. **d** Same map obtained using two months of noise after the Md3.3 earthquake (April 26th, 2020, circled number 2), when the transfer structure reappears.

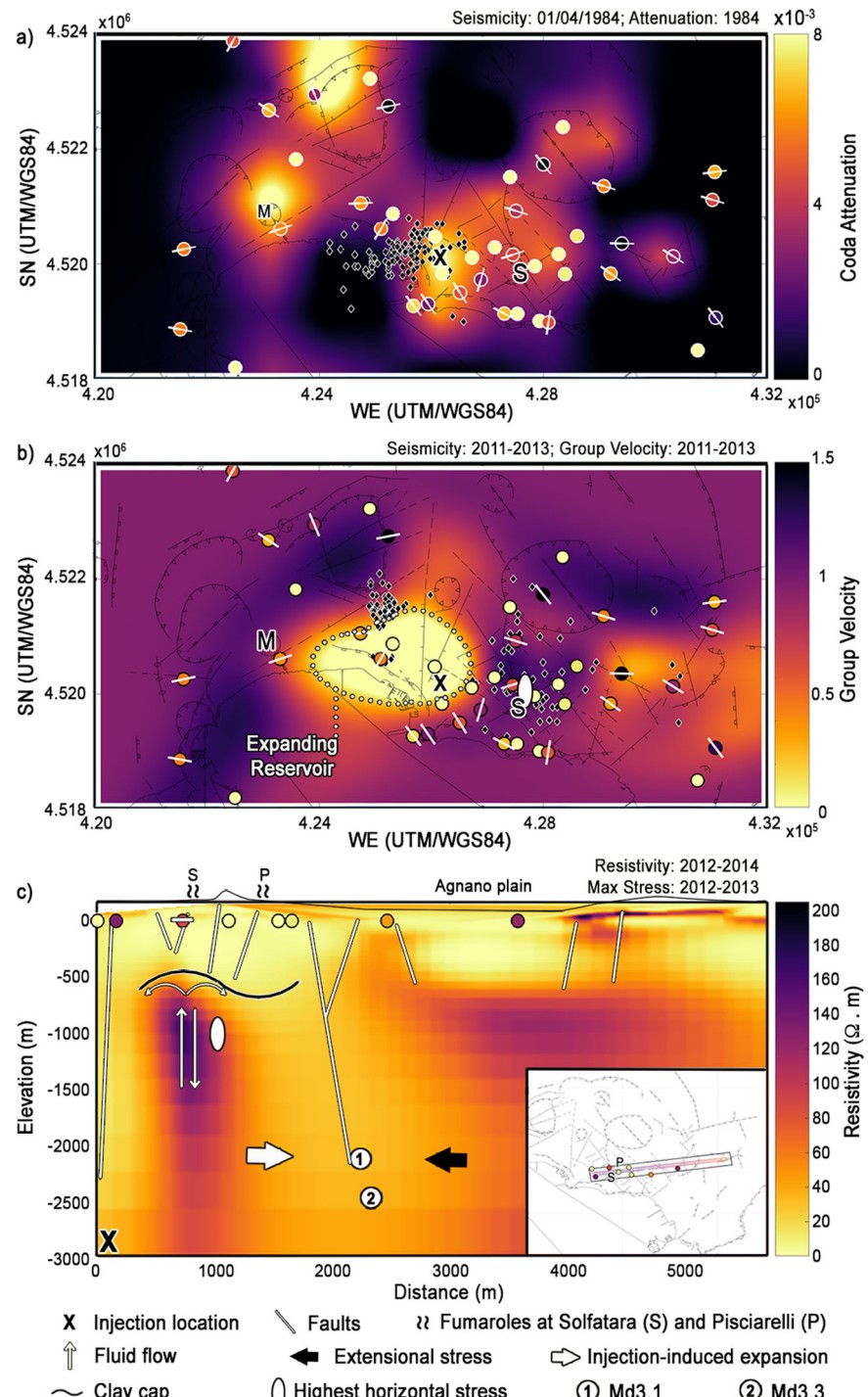

**Fig. 2 Comparison of polarization with velocity, attenuation, resistivity and stress. a** The low-frequency resultant lengths (colour-mapped in Fig. 1) and azimuths obtained at stations recording in 2009 and 2017 are compared with the high-attenuation signature (black cross, coda attenuation of 0.008[29]) of the injections that opened the low-velocity hydrothermal system in 1984[16]. The black diamonds show the earthquakes recorded on April 1st (maximum Md = 4.1)[16]. **b** In 2011-2013, a low-velocity aseismic reservoir was expanding from the injection location[6,35] (white dotted curve). The white ellipse at Solfatara is the point of highest lateral stress in 2011–2013[31]. The black diamonds show seismicity in the same period. **c** A resistive plume feeds the fumaroles at Solfatara and Pisciarelli. Faults and a clay cap[15] constrain the plume. The profile and nearby seismic stations are shown inside the rectangle in the lower inset. The thick white arrow refers to the east-directed expansion[20,31] from the deep injection point. The thick black arrow shows the west-directed extension of the caldera-bounding faults[24-26] that bind a resistive metasomatic reservoir[15] under the Agnano plain. The Md3.1 and Md3.3 earthquakes[38] nucleate on a deep fault within conductive liquid-bearing metasediments[15].

outgassing at the surface[28] (Fig. 1a, black dotted line). At higher frequencies (1–5 Hz, Methods, Supplementary Fig. 1), regional and caldera-bounding faults disappear due to the sensitivity of ambient noise to shallower and smaller structures[36]. High resultant lengths

and polarized azimuths mark the NW-SE-trending extensional faults with exceptional stability over years (Supplementary Figs. 2–3), months (Supplementary Figs. 4a–5), days (Supplementary Fig. 6), and even a single hour (Supplementary Fig. 4b) in

periods of low (2009 and 2017) and high (2018–2020) seismic releases (Methods). Instead, the transfer structure develops SW-NE, the hypothesized direction of the volcanic ridge[22] that connects paired sources of deformation under the caldera[27], only in periods of low seismic release (Fig. 1a). When hydrothermal pressure, gas emission, and seismicity increase (2018)[38,39], the transfer structure depolarizes, allowing to monitor fluid injections and migrations leading to high-duration-magnitude (Md > 3) earthquakes (Fig. 1b–d). The results and the transfer structure discussed in this paper are independent of mapping interpolation (Supplementary Fig. 7). The high-attenuation[29] signature of the repeated injections that caused the strongest volcano-tectonic event recorded at the caldera on April 1st, 1984 (Md = 4.1)[16] shows no polarization (is unpolarized) even in 2009 and 2017 (Fig. 2a). The transfer structure crosses the high-velocity eastern sector of the Solfatara crater[6] and borders the SN anomaly that accumulates the highest lateral stress and produces seismicity during unrest[31,35] (Fig. 2b, white ellipsoid).

**Imaging stressed fluid-filled structures.** In the eastern caldera, the highest resultant lengths correspond to azimuths consistently parallel to the NW-SE high-velocity extensional faults[15,22,23] (Fig. 1). The area is wide enough to become a high-velocity waveguide for horizontally-polarized isotropic S waves generated either in the centre of the Tyrrhenian Sea[2] or across the near coastline (Supplementary Fig. 8a, b). This waveguide could explain azimuths parallel to the trend for both source configurations at stations with $R > 0.25$ (Methods). Still, a far-field source[2] better fits azimuths observed across the entire caldera (Supplementary Fig. 8a, b, see the residuals). Far-field sources cannot explain azimuths perpendicular to the primary direction (SW-NE) of the transfer structure in 2009 and 2017 (Fig. 1a). These azimuths could be a consequence of seismic anisotropy, which tracks permanent directional signatures from the deep Earth mantle[40] to hydrated subducting slabs[41]. If low-velocity faults are wide enough, stiffness anisotropy[12,13] and trapping and reverberations[42] on high-dip fault walls can polarize noise perpendicular to fault walls. Across the transfer structure, azimuths indeed develop perpendicular to high-dip fault walls (Fig. 2c) and crack anisotropy at least at Solfatara[32]. However, the transfer structure is a small high-velocity structure[6] (Fig. 2b) consequence of lateral stress accumulated in the crust[20,21]. Azimuths across this structure better fit those obtained for sources generated at the near coastline[2] (Supplementary Fig. 8a, b, right). Near-field sources[28] seem a more likely controller of azimuths than anisotropy, even if anisotropy can increase polarization across similarly compressed structures[8,9].

Depolarization of the transfer structure explains stress release and structural changes in the volcano. While the trend marking extensional faults appears consistent over time, the transfer structure only polarizes during periods of lower seismic and geochemical release (2009 and 2017)[38,39], when deep injections and hydrothermal recharge are sparse and rarely coupled[39,43] (Fig. 1a). The structure is in contact with the high-attenuation[29] and deforming[29,31] location of deep injections (Fig. 2a, black cross). It runs along:

(1) the semi-circular east and north borders of a reservoir that was expanding in 2011–2013 (Fig. 2b);
(2) the lobe-shaped maxima of horizontal stresses observed using InSAR methods;[31]
(3) an abrupt structural variation in tidal tilting from WE to SW-NE[44].

These geophysical responses and maps are linked to the sub-caprock migration of over-saturated pressurized fluids[14] of hydrothermal[17,18,23] or magmatic[19,21,45] origin, which produce

persistent low-frequency noise and long-period events at the caldera[37]. The high-scattering fluids rising and migrating from deep injections pervade fractures, creating local noise that progressively intensifies[28] and depolarizes the transfer structure (Fig. 1b, c). In the presence of high-velocity contrasts[6], stations within one wavelength from such extended sources lose polarization in the heterogeneous medium (Supplementary Fig. 8a, b, right, $R$ decrease at station ACL2). This behaviour could be the cause of the depolarization of the transfer structure at Solfatara in 2018, when fluid injections and migrations acted as extended sources and connected the central and eastern reservoirs (Fig. 1b). Fluids eventually flow through metasediments[15] located between transfer and extensional structures. These high-attenuation[16,29] fluid-filled sediments reduce ambient noise directionality between 0.2 and 1 Hz through scattering[46] and are the most consistent unpolarized structure during the decade (Fig. 1).

In 2019–2020, the pre-seismic (Fig. 1c) and post-seismic (Fig. 1d) patterns show the progressive depolarization induced by fluids migrating from the injection location to:

(1) the eastern sector of the Solfatara and Pisciarelli vents (S, Fig. 1), where the geochemical unrests of the last fifteen years have been monitored;[19,28]
(2) Monte Nuovo (M, Fig. 1), the location of the last eruption at Campi Flegrei (M, 1538AD), where fluids migrate and stress increases during unrest[45].

The Solfatara and Pisciarelli vents emit from 2000 to 3000 tons/day of $CO_2$ in the atmosphere[28]. They have been consistently deforming toward the east in the last 20 years[47,48], moving along with seismicity from the injection location[35]. Joint interpretations of resistivity, geochemistry, and field data[15,25] detect the plume that feeds these vents, the surrounding altered metasediments, and the eastern extensional faults that bind low-density metasomatized rocks[15] (Fig. 2c). In the western portion of Fig. 2c, the transfer structure crosses the capped resistive plume that stores steam and gas, feeding fumaroles. Here, injections of fluids from depth[18,28] coupled with meteoric recharge[43,47,49] produce lateral stress[31], with fluids eventually permeating the liquid-bearing sediments[15]. Gas-bearing fluids over-pressurized the eastern caldera between 2011 and 2013[11] due to concurrent lateral expansion of the source region[31] and saturation of the reservoirs[19]. This caused the highest horizontal stress east of the Solfatara feeder (white ellipse[31], Fig. 2b, c). The depolarization of the transfer structure that started in 2018 (Fig. 1b) marks the process that led to the highest seismic release in thirty-six years at the caldera[38,39]. Fluid injections from depth coupled with progressive permeability increases from heavy rains[43,47,49] started the seismic sequence in December 2019[38]. Stresses on the high-dip fault east of Solfatara (Figs. 2c and 3) generated two high-magnitude volcano-tectonic events[38] after minor earthquake swarms:[50] an Md3.1 on December 6th, 2019 and an Md3.3 on April 26th, 2020 (white circles 1 and 2, Figs. 1c, d, 2c and 3a, b).

**Monitoring stress and fluid migrations.** The 2019–2020 seismic sequence is the effect of pressurization of the hydrothermal system[39] induced by lateral stress and fluid migrations, which horizontal noise polarization can monitor. For example, the mechanical weakening of the crust[5] and the corresponding depolarization of ambient noise cross-correlations[9] after the Tohoku earthquake detect the release of stress and upward fluid migration at volcanoes hundreds of kilometres afar. In a stressed geothermal environment like Campi Flegrei, these surges appear at sharp lateral discontinuities as caldera-bounding faults. In September 2012, fluid injections activated western caldera

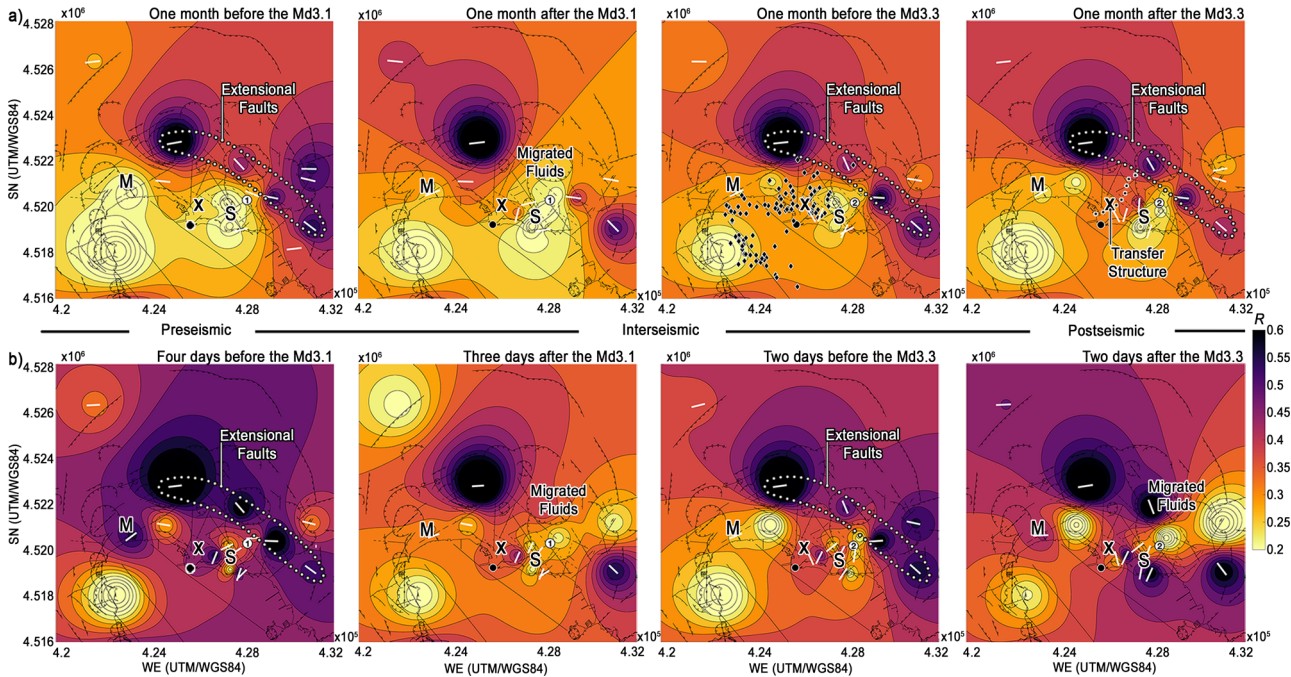

**Fig. 3 Build-up and release of fluid-induced stress. a** The polarization parameters have been plotted using data spread across one month before and after both the Md3.1 and the Md3.3 earthquakes. Black diamonds correspond to the earthquake locations between June and December 1984[16]. The white dotted line contours the extensional faults when visible and continuous. Migrated fluids break this continuity. **b** The olarization parameters were computed using three hours of noise on a single day, before and after the Md3.1 and Md3.3.

faults near Monte Nuovo (Fig. 2b, M, western black diamonds)[30]. The resultant lengths measured over months at the nearest station detect the permanent depolarization following the earthquakes, in analogy to interferometric analyses[8,9] (Methods, Supplementary Fig. 9). Fluid migrations between western and eastern caldera were the mechanism that released stress at the end of the 1984 unrest[11]. Months after the 2012 swarm, it is the part of the eastern caldera located between transfer and extensional structures (Fig. 1a) that suffered the highest long-lasting velocity reductions (>0.1%)[11]. These reductions indicate the area bearing the highest concentration of pressurized fluids[10,11], which is also the part of Campi Flegrei most likely to form new hydrothermal vents and nucleate earthquakes[51,52]. The temporal patterns (Figs. 1c, d, 3a, b, Supplementary Figs. 5, 6) clarify that fluid migrations connecting western and eastern caldera coexist and possibly drive stress build-up and release through the 2019–2020 seismic sequence. Fluids migrate under the Campi Flegrei caprock, which forbids surges directly above the primary source of deformation[14]. After each earthquake in 2019–20 (Supplementary Fig. 10), the change in polarization is equivalent to that observed after the earthquakes in 2012 (Supplementary Fig. 9). It is analogue to the decrease in ambient noise polarization caused by hydrothermal fluid surges at Mount Fuji after the Tohoku earthquake[9]. Unlike Mount Fuji, horizontal stress was already in a critical state at Campi Flegrei due to magma degassing[19] and supercritical fluids pressurized under the caprock[18,19,45].

During the pre-seismic period (Figs. 1c, 3a), after minor swarms stroke the eastern caldera[28], the unpolarized anomaly under the Solfatara and Pisciarelli vents developed north to south. After the Md3.1 earthquake, this anomaly expanded toward the eastern flank of the Solfatara and Pisciarelli vents (Fig. 3b), matching the hypothesized low-gravity fluid-ascension path between the two vents[25,33]. During the inter-seismic period, the anomalies in the western and eastern caldera connected across seismic pathways that released stress and closed the 1984 unrest[16]

(Fig. 3a, diamonds). These maps track fluids generated by the deformation source[19,27] and over-pressurized in the capped system[18,34]. The fluids migrated both seismically and aseismically in 2020[38], pressurizing the eastern hydrothermal system until the Md3.3 released stress. The Md3.3 sealed the migration by polarizing noise across the transfer structure (Fig. 3a, rightmost panel). By May-June 2020, the eastern unpolarized anomaly was one km east of its original location. It comprised the earthquake location (compare Fig. 3a, left to right) and an area polarized before the sequence (Fig. 1a-c). This dislocation is the seismic signature of the persistent lateral stress leading to fluid migrations toward the eastern caldera[35].

**Toward real-time monitoring with depolarized noise**. Heat increase and critical degassing pressure from depth[19] coupled with hydrothermal recharge[43] make the area between extensional faults and transfer structure (Fig. 1a, d) most likely to break in the future[51–53]. Knowing the delay between deep fluid injections and activation of faults in the eastern caldera allows us to investigate the real-time potential of depolarized noise. This delay was obtained by recent thermo-hydro-mechanical modelling[50]. Depending on injection volumes, fluids were injected at the base of faults in the east caldera between three and five days before the Md3.1. Fig. 3b and Supplementary Fig. 6 show polarization parameters measured using three hours of noise each day in these periods. After a consistent depolarization five days before the earthquake (Supplementary Fig. 6, 01/12/2019), the R increased at all the stations around the location of the Md3.1 (Fig. 3b, pre-seismic), in a manner that is consistent with an increase in compression preceding earthquakes[50]. After the Md3.1, the unpolarized anomaly east of Solfatara expanded toward the east (Fig. 3b) with significant statistical variations at stations in the eastern caldera (Supplementary Fig. 10). Similar maps are obtained in a shorter time interval (one to three days) around the Md3.3 to account for the increase in pore pressure following the inter-seismic period[50] (Fig. 3b, Supplementary

Fig. 6). Two days before the Md3.3, the eastern unpolarized anomaly had focused on the earthquake location. Two days after the Md3.3, fluids had outflown the area east of the Md3.3[15], depolarizing the eastern extensional trend like after the Md3.1 (Fig. 3b, from left to right). These spatial and temporal relations confirm that depolarized noise can monitor deep sub-caprock migrations of fluids preceding and following higher-magnitude earthquakes.

## Discussion

Ambient noise polarization answers the long-standing question of how fluids feed hydrothermal vents, building and releasing stress at Campi Flegrei. A transfer structure connects the central deforming caldera to regional extensional faults, running under a caprock whose characteristics allow over-pressurization, lateral fluid migration, and strong lateral deformation[14]. The area of major volcanic and seismic hazard[51,52] is sandwiched between transfer and extensional systems. The opening of the transfer structure detects deep fluid migrations toward the surface. These fluids trigger changes in polarization patterns[37] allowing mapping of stress build-up and release through further eastern fluid migrations. Both the caprock and associated high lateral stress at the caldera seem crucial for monitoring volcanoes with noise depolarization. Discriminating depolarization from processing uncertainties would be difficult without the persistent high polarization across the structures that bear most of this stress. While this could be an important limitation at volcanoes that release stress frequently, like Etna, and that present different lithological contrasts, the technique seems ideally suited to image and monitor volcanoes with long periods of repose.

Temporal scanning of depolarized noise represents a substantial step toward instantaneous imaging of hydrothermal expansion, leading to earthquakes in stressed calderas. The relationship between polarized noise and stressed structures provides a unique tool to constrain stress magnitudes and directions, the first step for a reliable physics-based vent forecasting[53]. Polarization measurements from ambient noise interferometry require yearly recordings for stable imaging, several days of monitoring measurements, and high amounts of processing. As previously hypothesized[9,12,13], horizontal noise polarization can achieve similar results using hours of noise and minimal processing.

## Methods

### Data processing and estimates of horizontal polarization values. The seismic noise recordings used in this study are obtained across eleven years from broadband stations[6,11,28,43] (Fig. 1a). They comprise:

1. Data for the first six months of 2017 was obtained from 17 mobile and 6 permanent broadband stations of the INGV, Sezione di Napoli-Osservatorio Vesuviano (INGV-OV) seismic network. The signal was extracted from the continuous six-month-long (January-June) recordings by choosing one week/month and 1 hr/day (00:00-01:00 UTC) of each week: an amount of about 42 h of seismic noise per station. Samples of noise recorded during night-time were chosen to minimize spurious sources caused by anthropic activity[1].

2. Data recorded in 2009 by 20 temporary stations installed during the Unrest seismic campaign[37], and by 4 additional broadband stations (3 mobile and 1 permanent installation) that were in operation in 2009 but no longer in 2017. In this case, due to the short period of acquisition (the Unrest campaign lasted from 9 to 26 March), we extracted samples of three hours (00:00-03:00 UTC) from the continuous recordings performed during the experiment obtaining (on average) about 45 h of signal/station.
The complete data set of 2009 and 2017 comprises a total of 47 sites (Fig. 1a).

3. Data randomly sampled in the first six months of 2018 (recorded between 00:00-03:00 UTC) at 23 broadband stations of the mobile and permanent networks of the INGV-OV. We extracted about 48 h of seismic noise per station. In addition, we used 1 h (of this dataset) at all stations to demonstrate the hourly stability of the patterns across the extensional trend (Fig. 1b, Supplementary Fig. 4).

4. Data recorded in 2019 (September-December) and 2020 (January-June) at a higher sampling level to test the monitoring potential before and after earthquakes (Figs. 1c, d and 3). The samples were extracted after selecting 9 days/month, except in December 2019 and April 2020. During these months, we selected 12 days to sample periods immediately before and after the earthquakes. For each day, we always select the same 3 h (01:00–04:00 UTC). We obtained 117 h (for 2019) and 171 h (for 2020) of signal/station at 20 broadband stations of the mobile and permanent network of the INGV-OV seismic network.

The seismic noise samples were filtered by applying an a-causal Butterworth filter in the bands 0.2–1 Hz and 1–5 Hz. Resultant lengths ($R$) and azimuths of the seismic wavefield were obtained by applying the covariance matrix method[12,13,36,37] to three-component seismograms at each station, using contiguous sliding windows containing three-wave cycles of the maximum period. $R$ ranges in the interval [0,1]. The closer it is to one, the more concentrated the values around the mean polarization direction are. Data for which the rectilinearity[12,13] was less than 0.5 were discarded, as the angular parameters are associated with seismic wave propagation only if above this threshold[36]. We focused on horizontal ground motion polarization as it is strongly controlled by the medium properties (e.g., presence of faults and cracks)[12,13]. We thus selected the azimuth values associated with a high horizontal polarization degree, fixing an incidence angle of 45° as threshold[12]. Supplementary Fig. 1a, b shows $R$ and azimuths measured at each station for 2009 and 2017. Panels c and d show the corresponding interpolated mapping. Compared to 0.2–1 Hz, the 1–5 Hz patterns (Supplementary Fig. 1b, d) are more affected by anthropic noise[1]. The Matlab© data processing software necessary to obtain polarization parameters is available at the Open Science Framework link provided in the Code Availability section.

### Stability of the polarization through time. We compared the results evaluated at five stations (ASBG, CELG, CMSA, CSOB, OMN2 OVDG) of the permanent and mobile networks that were operative in 2009 and 2017. In none of these cases, variations of the polarization features were observed (Supplementary Fig. 2a). A bootstrap test calculated 1000 means of random samples drawn from the $R$ distribution. The subtraction of the average $R$ of the real distribution and the bootstrap mean (Supplementary Fig. 2b) shows that, over 47 stations recording in these periods, 41 present minimal changes in $R$ ( < 0.1).

### Stability of the polarization patterns measured during 2017 and 2018.
We assess the stability of our results when using data recorded over six months for 2017 and 2018 (Supplementary Fig. 3, blue and orange lines, respectively). A total of 22 stations recorded noise in both periods. The parameters are compared with one hour of a signal recorded simultaneously at all stations in 2018 (Supplementary Fig. 3, green, this was possible only for 20 stations). In the figure, there is a 180° periodicity so that apparent changes in azimuths like that at station RENG are uninfluential. Azimuths show minimal differences for $R > 0.3$ and are always within uncertainties, while $R$ values are most stable across the extensional trend (red labelled, Supplementary Fig. 3). The comparison between patterns computed over 6 months and 1 h in 2018 is reported in Supplementary Fig. 4. When considering a single hour, minimal variations are observed across the extensional trend.

### Monthly and daily variations during seismic unrest. Supplementary Fig. 5 shows the monthly variation in the polarization patterns between September 2019 and June 2020. Monthly variations of $R$ and azimuth mean values for the pre-seismic period (September-November 2019) show a progressive increase of $R$ at all stations. After the Md3.1 earthquake (circled number **1**, December 2019-January 2020) the eastern unpolarized anomaly moves to comprise the earthquake location, while the western caldera polarizes. In the inter-seismic period (January-March 2020) western and eastern unpolarized anomalies connect the north of the deformation source while the eastern unpolarized anomaly moves back to its original location. The Md3.3 post-seismic maps (circled number **2**, May-June 2020) show polarization increases in the sealed central migration system (June 2020), while the eastern unpolarized anomaly moves to the earthquake location. Supplementary Fig. 6 shows the daily variations.

### Simulation of isotropic homogeneous horizontal noise polarization. We model noise polarization from an extended line of noise sources located in the central Tyrrhenian basin and from a circle representing noise sources offshore[2]. As sources, we use Morlet wavelets of dominant frequency 0.7 Hz, repeating every 8 s in an isotropic simulation of the wave-equation. The staggered stress-displacement description of SH propagation incorporates viscoelasticity from the available total attenuation model[16] by using memory variables assuming constant-Q Zener model[54]. To obtain seismic velocities from displacements, we apply a finite-impulse-differentiator filter of order 24. The propagation grid extends to the area shown in Fig. 1a (Supplementary Table 1). The strains are obtained from their relationship with displacements, using a spatial derivative operator of fourth-order. The discretization of the memory-variable equations is performed using the central differences operator for the time derivative and the mean value operator for the memory variable. Two sponges attenuate boundary propagation.

The finite-difference simulations are most unstable if polarization azimuths are either 0° or 90°[54] and near the center of the caldera for circular polarization. We used grid spacings of 40 m for the two source settings, obtaining $R$ varying in the intervals shown in Supplementary Fig. 8a, b. Thus, the simulation grid comprises 750 nodes regularly spaced at 40 m; of these, 150 nodes on each side are allocated for the absorption boundary conditions[16]. The lowest/highest velocities[16] used are 0.5 km/s and 1.5 km/s. For S waves of velocity $v_S$ and a grid step $\Delta l$, stability is given at times of at least $\Delta t = \frac{6}{7\sqrt{3}} \frac{\Delta l}{v_S} = \frac{6}{7\sqrt{3}} \frac{40}{1.5 \times 10^3} = 12$ ms in an isotropic medium[54]. To consider the variations induced by anelasticity and grid dispersion we reduced the time step to $\Delta t{=}1$ ms for noise signals lasting 100 s.

We simulated seismograms at all stations recording noise in 2009 and 2017 and having a minimum $R{=}0.25$ in the results (Supplementary Fig. 8a, b). The decrease in the homogeneous cases (Supplementary Fig. 8a) is due to numerical instability, the finiteness of the differences, and boundary conditions only[54]. Depolarization (reductions of $R$) in the homogeneous case and at these frequencies are minimal (below 1%) at all stations for both source configurations (Supplementary Fig. 8a). The polarization parameters are retrieved with a blind test, where synthetic seismograms are processed without inputs on the original source polarization. The results for the homogeneous cases are shown in panel a) and are compared with real azimuths in panel c. The square residuals between azimuths in the two source configurations indicate that a far-field source is on average more likely to reproduce results (a line residual of 208 against 294). This gives us a threshold to interpret if the sole existence of velocity contrasts can reduce $R$ at the levels observed in the data.

**Simulation of isotropic heterogeneous horizontal noise polarization**. The results of the polarization analysis (Fig. 1a) are inserted in the propagation matrix with a 50% increase in shear modulus, a value derived by ambient noise tomography[16] and fixing constant density values[20]. The change is applied only to nodes where $R > 0.31$ (Supplementary Fig. 8b), as 0.31 is the average $R$-value over the 2009 and 2017 datasets. For the extensional path, we restricted the area of change to within the extensional faults. The results of the blind tests show a strong reduction of $R$ at station ACL2 ($R{\sim}0.5$), the only station both inside the waveguide and within one wavelength from noise sources. Without waveguide and with the same source configuration, no near-field trapped and scattered wave responsible for decreasing polarization can develop. This explains lower $R$ values as due to a combination of medium heterogeneity and extended near-field sources.

The azimuths slightly rotate parallel to the extensional trend (NW-SE) in the eastern caldera independently of the starting source polarization (Supplementary Fig. 8b); yet only near-field coastline sources reproduce azimuths perpendicular to the primary direction of the transfer structure. The lowest residuals are produced by the heterogeneous case with far line sources (residuals of 202 against 295). However, at least for the simulated isotropic case for this frequency band, the sole existence of high-velocity heterogeneity as observed at Campi Flegrei has only minor effects on azimuths: these are primarily controlled by the location of noise sources.

**Changes of horizontal noise polarization with swarms - 2012**. High frequency (1–5 Hz) horizontal noise loses polarization ($R$ strongly decreases) permanently near the location of the last eruption of the volcano (Monte Nuovo, 1538 AD) after September 2012 swarm[30,38], which was one of the strongest recorded at the caldera between 1984 and 2019 (Supplementary Fig. 9, right). An unequal variance t-test on the resultant length calculated between March 2012 and January 2013, confirmed ($p < 0.05$) the hypothesis that the two sample populations (before and after September 2012) have different means. The permanent decrease of $R$ is the likely consequence of fluids that permeated the area, saturating and isotropizing the system[9]. Between 0.2 Hz and 1 Hz (Supplementary Fig. 9, left) the hypothesis of the unequal mean is confirmed only considering data between June 2012 and January 2013. The cause is a small swarm in April 2012[30,38] that decreases $R$ temporarily. After this swarm, we observe a progressive low-frequency increase of $R$, indicative of pressurization of the deeper systems.

## Data availability

The raw ambient noise data are available under restricted access as they are collected by the INGV, Sezione di Napoli - Osservatorio Vesuviano on behalf of the Italian Civil Protection. Access to these data can be obtained by contacting the corresponding author, who will request approval from the director of the INGV, Sezione di Napoli - Osservatorio Vesuviano. The data generated in this study and necessary to reproduce figures have been deposited in the Open Science Framework database under accession code[55].

## Code availability

Raw data were transformed into SAC format and processed using scripts developed using Matlab© version 2019a to obtain the polarization parameters. The data processing scripts, resulting files, and codes necessary to (1) process SAC data, (2) create the figures in the main text, and (3) perform wave-equation modelling have been deposited in the Open Science Framework[55]. The wave-equation modelling scripts are available as a release of the corresponding GitHub project[56]. Supplementary Figs. 9–10 have been drawn using Golden Software Grapher™. The final figure layouts were prepared using Photoshop CS©.

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

## Acknowledgements

We thank the staff at the INGV-Sezione di Napoli, Osservatorio Vesuviano, for providing compiled seismic catalogues, the routine locations of volcano-tectonic earthquakes, and access to real-time data. Giuseppe Vilardo and Agata Siniscalchi provided the shapefiles used to plot faults and fractures and the resistivity model. TeMaS - Terrestrial Magmatic Systems Research Area of the Johannes Gutenberg University (Landesinitiative des Landes Rheinland-Pfalz) has funded L.D.S.

## Author contributions

S.P. conceived the initial idea to use the resultant length of polarization vector as a tool to image the medium properties, analyzed all seismic data, and performed all the measurements of seismic polarization from ambient noise through years, months, and daily analyses. L. D. S. performed the wave-equation modelling, created the tools for the generation of Figures to interpret polarization with existing geophysical models, and wrote the first draft of the paper. The authors completed the manuscript together.

## Funding

## Competing interests

The authors declare no competing interests.
