## [Peer Review File · Nature Communications]

REVIEWER COMMENTS

Reviewer #1 (Remarks to the Author):

The paper by Petrosino and DeSiena investigates the spatial and temporal resolution of polarized noise fluids migrating through the Campi Flegrei caldera. The subject matter of the paper is certainly interesting, original, and appropriate to be published in a scientific journal. However, it is very difficult to determine this science and the results are presented.

The main concern regards the general presentation and structure of the paper that prevents from capturing flaws in the data analysis, separate data from interpretation, and finally conclusions. What I found missing is a roadmap for the reader and clear paragraph structure. There is also a lot of technical jargon that I found difficult to read through. I strongly suggest editing for language, clarity and writing style before acceptance.

Here below some examples:

General comment: I would invite the authors to briefly clarify upfront the term "Polarized noise" so lay people can read and understand the main results.

Lines 35-36: Why "hazardous CO₂-bearing fluids"? CO₂ is not a hazardous fluid. Do you mean "pressurized"? I would also encourage the authors to go back to the source to find the material that they have quoted. For example, cite original papers imaging the presence of pressurized fluids – Vanorio et al., 2005, JGR. Then, Battaglia et al., 2008 Geophys. Prospect. Avoid lumping references such as [xx-yy-zz]. There are many examples throughout the paper. Instead, summarize the main contribution of each referenced paper in a separate sentence and by including the reference number.

Line 39: "Polarized noise can see through the overlying rocks to catch these marks." This sentence sounds awkward. Please, re-phase.

Line 65-66 Where is the transfer structure in the figure the text is referring to? It would be helpful if the authors added features to Figure 2b so as to accompany the reader throughout the text.

Line 67 "identify a SN anomaly that accumulates the highest lateral stress during unrest"...where is the SN anomaly in the figure?

Line 109 "Depolarization of the 2009-2017 transfer structure...". What does "depolarization of a transfer structure mean"? The authors should try to explain the used terminology, or avoid jargon that unnecessarily complicates language, rather than informing the audience.

Line 113 "...location of deep injections (Fig. 2a, black cross)". How was the location of this injection estimated?

Line 122 "In the presence of high-velocity contrasts". What does characterize the high-velocity contrasts? Is this contrast lithological or structural? Velocity log data and tomographic images show the presence of smooth velocity variations. In particular, the lithological description of the formations crossed by the wells in the Campi Flegrei caldera shows the caldera infill being characterized by the presence of tuffs and tuffites having different cementation and/or alteration degree.

Line 124 "when the central and eastern unpolarized reservoirs connect..". This sentence sounds awkward. What's an unpolarized reservoir? Please, re-phase.

Line 125 – "Fluids eventually outflow on metasediments". Check English usage.

Lines 129-130 "The pre-seismic (Fig. 1c) and post-seismic (Fig. 1d) patterns show the progressive depolarization induced by fluids migrating from the injection location to..." How were pre-seismic patterns

obtained?

Reviewer #2 (Remarks to the Author):

Dear Dr. De Siena and Dr. Petrosino,

Very interesting manuscript. Your article "Fluid migrations and volcanic earthquakes from depolarized ambient noise" studies how noise polarization can be used as a new tool to monitor pressurized fluids in volcanic systems, which may induce volcano-tectonic events and migration after seismic sequences. This technique can also be used for monthly and daily monitoring of fluid migration with minimum processing. The test case is focused on the Campi Flegrei caldera (South Italy), one of the most instrumented and long-term studied areas. Using low-frequency ambient noise (0.2-1 Hz), you found that the caldera-bounding extensional faults and a transfer structure crossing the deforming center and vents are consistently polarized during periods of low seismicity. When the seismic and geochemical activity increases, these structures, in particular the transfer feature, depolarize due to the migration of pressurized fluids which may trigger larger magnitude volcano-tectonic events (2019 and 2020 sequences).

Your paper presents a unique and novel approach to improve our understanding of volcanic eruptions forecasting and monitor volcanic unrest, as well as short-term monitoring of pressurized fluids. I think the method you propose here may also help the entire community to better understand volcano-earthquake triggering. Finally, I think this approach may be used in combination with other volcanic hazard forecasting tools, such as geochemical, remote sensing and other geophysical data.

I provide some comments and suggestions on the content, wording and presentation below, which I hope you will find helpful. For simplicity, I abbreviated the number of lines as e.g., L1 means Line #1:

1. I am not sure to understand how exactly fluid migrations are monitored if the transfer structure depolarizes (L52). In Fig 1b-d I don't clearly see the fluid geometry, unless the fluids are only present where the value of R becomes low. Although I don't doubt that the polarization of both the transfer structure and in particular the extensional caldera-bounding faults are pretty consistent, from Fig. 1 I am not able to track how these fluids migrate to the east triggering the larger-magnitude earthquakes (circled number 1 and 2 in Fig. 1c and d). Maybe, further explanation may be needed.
2. L55-L57: it is not clear what the authors mean, since data from 1984 are plotted against data of 2009-2017. Therefore, I am not sure if there is a straight connection/correlation between the two datasets. I would reword the sentence as: "The high-attenuation signature of ... seems to coincide with the area of low polarization, still suggesting actual presence of pressurized fluids within the same region (Fig. 2a)". If, instead, the authors suggest a sort of correlations between the actual unpolarized anomaly and the high attenuation of three decades ago, my questions would be: what may be the physical explanation or evidence to justify a temporal correlation between them?
3. L65-66: the high-velocity/stress barrier doesn't exactly coincide with the transfer structure of Fig. 1a. In Fig. 2b we can see many low R areas in the studied region. It can be a smaller scale as shown in Fig. 2b with perpendicular polarization to the faults. I would suggest to say that the transfer area is located in the area of high velocities and stresses. Otherwise, how would they explain the low depolarization calculations (yellow dots in Fig. 2b) within the high-velocity structure?
4. In Fig. 1a it is not clear whether the analyzed data are collected in the period between 2009 and 2017 or only 2009 and 2017. In L99 it says in between, while in the caption of Fig. 1 and 2 it seems the data are two distinct datasets, only from 2009 and 2017, respectively.
5. L119-120: I think the authors may remove the second part of the sentence "..., adding persistent low-frequency noise and long-period events". The reason is that it makes confusion to the reader with the next

sentence. If the migration of pressurized fluid tends to depolarize the structure, it indicates that their signals can cause high-frequency noise, not long-periods events. Otherwise, I would reword the sentence as: "..., adding high-frequency components to a background low-frequency noise and long-period events".

6. L124: I would change "apparent" with "seems to occur", since apparent can have two meanings, one as a synonym of "clear/obvious" and the other as "maybe true, but not necessarily". I am not sure which version of the term the authors mean.

7. L125: I am a bit surprised about the role of the metasediments. While I may understand how sediments may reduce ambient noise, I am not sure whether these sediments are present only in the last decade (Fig. 1) and not present before (I would think they were also there three decades before). One thing it would be very interesting to look at is the distribution of lithotypes in the geologic map of the area and compare it with the polarization results, in order to confirm whether or not there is a spatial correlation between structures and lithology (metasediments). Only in this case, this cause-effect would be consistent.

8. L140-146: I would re-organize this paragraph as follows: "Gas-bearing fluids over-pressurized the eastern caldera between 2011 and 2013 due to concurrent lateral expansion of the source region and saturation of the reservoirs. This caused the highest horizontal stress in the area east of the Solfatara feeder (white diamond, Fig. 2b,c). The depolarization of the transfer structure that started in 2018 (Fig. 1b) led to the highest seismic release in thirty-six years at the caldera. Here, fluid injections from depth coupled with progressive permeability increases from heavy rains started the seismic sequence in December 2019". In this way, it seems to appear a temporal cause-effect.

9. L152: I would put: "... which horizontal noise polarization can monitor. For example, the mechanical weakening ...". Otherwise, the reader may wonder what the correlation between Campi Flegrei and the Tohoku-oki earthquake is.

10. L163-165: I would rewrite this sentence because it misses some subjects. I am not sure what are the subjects of "mostly likely to erupt, form new hydrothermal vents, and nucleate earthquakes". Are the velocity reductions? Fluids? Area? Moreover, I am not a fan of the word "symptomatic". You may use another word like "denote/indicate".

11. L182: "sealed the migration...".

12. L195: I would change "Once informed by ..." with: "Through thermo-...".

13. Extended Data Figure 1: Just a general comment/suggestion. Although I know that the number of stations is limited and concentrated in a small and narrow region, the squared interpolation from each station can cause not realistic artefacts, in particular in regions where there are no data (for example, in the north-east, west, south, and south-east of the studied region). Despite the fact that in the high-station region the interpolation may be close to the pointwise measurement, I would suggest removing the interpolation in areas where there are no data.

14. L272 (caption): I think there is a typo. It is not "b)" but "c-d)".

15. L275-277: Again, it is unclear if the authors mean between 2009 and 2017, or the two distinct years.

16. Extended Data Figure 4: it misses the colorbar. Personally, I think that this plot is very interesting since it corroborates the hypothesis to use noise depolarization for real-time monitoring.

17. L309: I think there is a typo. It should be "January-March 2020" not 2019 (at least from the plots below).

18. L313-314: I don't understand the following sentence: "..., where the intervals have been interpreted using the results of thermo-hydro-mechanical modelling". I would suggest removing this sentence or provide

further information regarding the choice of the time intervals.

19. L339: Again, it is unclear if the authors mean between 2009 and 2017, or the two distinct years.

20. L340-342: I don't quite understand these two sentences. The decrease of what? R ? Moreover, what is the lowest? R ? Because in Fig. 7a I can see in the brackets that the lowest R seems to occur for the computations where the sources are located at the coastlines, not in the far-field (0.9942 and 0.9983, respectively).

21. L343: I figured out that "L.D.S." and "S.P." refer to the name of the authors, but only after reading the "Author contributions". Personally, I would remove the names and make more an informal description of the method, something like: "The polarization parameters are retrieved with a blind test, where synthetic seismograms are processed without inputs on the original polarization".

22. L351: I wonder why the authors choose to increase 50% of shear modulus only in the regions of $R > 0.31$. Where does the choice of $R = 0.31$ come from? How does the result change if we choose $R > 0.4$? What may be a physical justification of 0.31?

23. Extended Data Figure 7: Since the residuals are pretty high and very similar between the homogeneous and heterogeneous case, I wonder how much panel 7b (heterogeneous case) adds to the paper. Visually comparing the two tests, I cannot see a very clear difference between them to justify the need of making the model more complex. The residuals do not improve much. Personally, I would leave the panels 7b out and comment in the caption that reasonable changes in shear modulus locally do not significantly improve the results/conclusions with respect to the homogeneous case.

24. L380-383: I am not completely sure about that at low-frequencies (0.2-1 Hz) we can see a clear decrease or increase of R . From the left panel (second plot from the top) the values of R do not clearly deviate. Although there are local decreases and increases (e.g., after the small swarm at Pozzuoli), there is not a very consistent and evident deviation as seen in the panel on the right (1-5 Hz). In this case, I agree that there is a permanent decrease of R after the Monte Nuovo swarm.

25. Extended Data Figure 8: There are two vertical lines in the left panel (0.2-1 Hz). What do they refer to? They are not defined either in the caption or in the text.

26. Extended Data Figure 9: I would put the labels (a and b) to the plot. In L390 I am not sure what "... and two and ..." refers to? In addition, the last sentence: "The variations indicate that the parameter is sensitive to the fluids released by the earthquakes" may make the reader wondering why and at what degree the parameter change is sensitive to the fluids, in particular those released by the seismic sequences. Which type of fluids do the author refer? At least, it is not completely clear to me. Further explanation may be needed.

Hope these suggestions will help to improve the manuscript. I look forward to seeing this published.

Best wishes,
Simone Puel

Reviewer #3 (Remarks to the Author):

Dear Authors,

I've read your paper "Fluid migrations and volcanic earthquakes from depolarized ambient noise" with great interest. I find your method beautiful in its simplicity, allowing to track migrations in one of the most volcano-hazardous regions in the world.

The manuscript is well organized and written, and easy to follow.

My main remark concerns the figures, which are difficult to read at first sight. I think the authors should define a maximum distance from a station their "squared interpolation" is no longer trustable, and mask those areas, this would give highlight where the map is really supported by data and where not.

I think the article lacks a small discussion on how this polarization analysis could be useful (or not) for other locations, where caprock is (or not) present?

Small remarks:

- Figure 2c: what is the source of the Resistivity data?
- Figure 2c: what are the uncertainties on the event locations?
- L200: "depolarization": does it mean "absence of significant polarization"? from "de-" it feels like it was polarized before, then got DE-polarized ?
- L216: not sure "compressed" is the best word here, do you mean something like "sandwiched"? or "located" ?
- L219: "instantaneous": from the method and the results shown, it seems indeed possible to have real time information
- L224: "minimal processing": with the comment for L219: I think it'd be worth quantifying the time/processing needed indeed.
- L232: "GMT" -> "UTC"
- L252: 117h vs 171h: the authors love to play with similar stuff (Md3.1 & Md3.3 too) :-)
- L254-259: It's unclear to me if the authors actually use the vertical component in the covariance matrix method, and if yes, does the incidence angle of the dominant polarization provide any interesting information?
- L307: "circled number 1" doesn't refer to the Figure where it's drawn
- L343: weird location to put the authors' contributions
- Extended figure 8: what is the source of the events' location ? The key changes should be highlighted on the top two panels (circle where the changes occur, draw trend lines before-after the MN Swarm)

"Data Analysis software" can be asked to LDS : why not putting it on the author's github?
Please cite all the software used to read/process/plot data, including the libraries & GIS software.

Looking forward to seeing this article published,

Best regards,
Thomas Lecocq

REVIEWER COMMENTS

Reviewer #1 (Remarks to the Author):

The paper by Petrosino and DeSiena investigates the spatial and temporal resolution of polarized noise fluids migrating through the Campi Flegrei caldera. The subject matter of the paper is certainly interesting, original, and appropriate to be published in a scientific journal. However, it is very difficult to determine this the science and the results are presented.

We thank you for the appreciation of our technical work and comments focused on improving readability and presentation. We are grateful for the remark about originality, which we consider the main strength of the paper. In the following, we address the concerns point by point, providing line numbers (L) corresponding to changes.

The main concern regards the general presentation and structure of the paper that prevents from capturing flaws in the data analysis, separate data from interpretation, and finally conclusions. What I found missing is a roadmap for the reader and clear paragraph structure.

The paper has been restructured in a more standard Introduction/Results/Discussion format. The required roadmap is provided at the end of the introduction, Lines (L) 67-74

There is also a lot of technical jargon that I found difficult to read through. I strongly suggest editing for language, clarity and writing style before acceptance.

Apart from the sentences highlighted below, we have gone through the paper to correct sentences and terms that might be challenging for the wider Earth Science community. For example, we have avoided the use of the term *unpolarized* (showing no polarization through time) where it could trade-off with *depolarized* (losing polarization). We specified the meaning of the terms we still deem necessary the first time they are mentioned (e.g., L16 for *depolarizations* and L94 for *unpolarized*).

Here below some examples:

General comment: I would invite the authors to briefly clarify upfront the term “Polarized noise” so lay people can read and understand the main results.

The paper introduces the concept of “*depolarized* noise” – noise that showed polarization across given Earth structures and directions and loses said polarization once pressurized fluids interact with the structures. This concept is now better explained from the abstract (L16) and at L30-37, as is the relatively straightforward term “polarized noise”.

Lines 35-36: Why “hazardous CO₂-bearing fluids”? CO₂ is not a hazardous fluid. Do you mean “pressurized”? I would also encourage the authors to go back to the source to find the material that they have quoted. For example, cite original papers imaging the presence of pressurized fluids – Vanorio et al., 2005, JGR. Then, Battaglia et al., 2008 Geophys. Prospect.

We changed the text to “pressurized fluids” (L41) and added the reference to Vanorio et al. 2005 (V05) directly after the mentioned sentence (L42, reference 18). Battaglia et al. 2008 (B08) offer the most comprehensive velocity model of the caldera, adding to the dataset of V05 data from the SERAPIS offshore seismic experiments; however, the onshore part of the velocity model depends primarily on the original dataset of V05. As V05 was the seminal paper proposing fluid-driven unrest, we believe this reference is sufficient for the paper’s need. Not citing this paper in the first version was

an oversight, as the concepts brought forward are not equivalent to those in Vanorio & Kanitpanyacharoen (2015). B08 is cited when mentioning the studies that image caldera-bounding faultss (L45, reference 23). Additional seminal papers are now included and highlighted in the References.

Avoid lumping references such as [xx-yy-zz]. There are many examples throughout the paper. Instead, summarize the main contribution of each referenced paper in a separate sentence and by including the reference number.

We revised the papers to avoid lumping references together. There are some cases where this was not possible. For example, references 35, 36, and 37 all discuss the presence of stress-bearing caldera-bounding faults and contribute to the faults shown in Figs. 1-3. However, Acocella et al. 1999 (34) was the only one using analogue modelling with fieldwork to state the existence of a transfer structure. We thus separated them at L44-50. The reviewer can find other examples highlighted in the text.

Given the format of the journal, we sometimes kept citation of a paper within the sentence where it is necessary, without having to write a sentence for each citation. For example, at L54-57 we write:

“The injection location was estimated from the source characteristics and spatial relation of repeated vertically-aligned earthquakes (black diamonds, Fig. 2a)¹⁶, whose timing was compared with geochemical data and the results of fluid and heat flow modelling¹⁹.”

We believe it is clear that reference 16 is the one where the vertically-aligned earthquakes are obtained, while fluid and heat flow modelling was performed by reference 19.

When the reference had to be explained to understand our analyses, we wrote a sentence for it, as at L227-229 for reference 50:

“Knowing the delay between deep fluid injections and activation of faults in the eastern caldera allows us to investigate the real-time potential of depolarized noise. This delay was obtained by recent thermo-hydro-mechanical modelling⁵⁰.”

Line 39: “Polarized noise can see through the overlying rocks to catch these marks.” This sentence sounds awkward. Please, re-phase.

This sentence has been removed. The concept is now expressed more properly in the introduction roadmap: “Our results show that polarized noise detects both the extensional faults and the transfer structure at the caldera during periods of low seismic release. The depolarization of the transfer structure marks both injections at the start of seismic unrest and lateral fluid migrations leading to earthquakes.” L70-72

Line 65-66 Where is the transfer structure in the figure the text is referring to? It would be helpful if the authors added features to Figure 2b so as to accompany the reader throughout the text.

We had a previous version of the paper with marked-up names on each feature, as the transfer structure and the extensional faults. See Extended Data Fig. 1c,d for an example. However, the figures resulted crowded if we had more than a couple of names in each figure (as it would be in Fig. 1). The guidelines of the Nature journal advise against imposing names on figure panels to avoid cluttering. They state (see <https://www.nature.com/nature/for-authors/formatting-guide>):

Where possible, text, including keys to symbols, should be provided in the legend rather than on the figure itself.

We believe this is good advice, given that the images could be even smaller in the final printed version. To find a compromise, we added letters (*TS* for transfer structure, *EF* for extensional faults, *IMF* for injected and migrating fluids, *ER* for expanding reservoir, and *MF* for migrated fluids) to accompany the reader, also following the advice of Reviewer 2. This lettering is explained in the captions of Figs. 1-3.

We leave the decision to reviewers and editor. It is easy for us to recreate figures with full names, like in Extended Data Fig. 1c,d.

Line 67 “identify a SN anomaly that accumulates the highest lateral stress during unrest”...where is the SN anomaly in the figure?

In the previous version, the structure was marked by a white diamond based on the InSAR results of Pepe et al. 2019, Remote Sens. and Env, which agree with the results of shear-wave splitting and gravity gradiometry. The symbol has been changed with a bigger ellipse with major axis SN under the eastern sector of Solfatara in Fig. 2b,c.

Line 109 “Depolarization of the 2009-2017 transfer structure...”. What does “depolarization of a transfer structure mean?” The authors should try to explain the used terminology, or avoid jargon that unnecessarily complicates language, rather than informing the audience.

As defined in the introduction, the term depolarization means “loss of polarization” of the structure through time, L35-37. We discussed before our changes to avoid jargon.

Line 113 “...location of deep injections (Fig. 2a, black cross)”. How was the location of this injection estimated?

The sentence necessary to understand how we infer the location are now written in the introduction at L52-58. De Siena et al. 2017, SR and De Siena et al. 2017, GRL use earthquake locations and coda attenuation tomography and find that the area of maximum attenuation (Fig. 2a) corresponds to the location of repeated vertically-aligned seismic clusters. The timing of these clusters corresponds to that of injections modelled by Chiodini et al. 2017 Nature: Communication with THOUGH simulations from a deep source of magmatic fluids. In De Siena et al. 2017, SR, the spatial relation between V_p/V_s (from V05), attenuation, seismic locations and source directivity is extensively discussed.

Line 122 “In the presence of high-velocity contrasts”. What does characterize the high-velocity contrasts? Is this contrast lithological or structural? Velocity log data and tomographic images show the presence of smooth velocity variations. In particular, the lithological description of the formations crossed by the wells in the Campi Flegrei caldera shows the caldera infill being characterized by the presence of tuffs and tuffites having different cementation and/or alteration degree.

We disagree that velocity tomography images represent smooth velocity variations, as the existence of a gradual, mostly-vertically-heterogeneous starting model is a requirement of the technique, and damping and smoothing are used through inversion, adding smoothness to the final model. Even so, the lateral velocity variations of these models (V05, as well as the more recent model of Calò and Tramelli, 2018) can be of the order of 10%, and velocity ratios span from 1.3 to 2.6, which is quite a significant variation. Calò and Tramelli (2018) find even more significant contrasts using the same datasets of V05 and B08 but a different tomographic technique. Using ambient noise surface waves (so

mapping the first 2 km of the volcano) and for the period 2011-2013 (which is nearer to our investigated period), ambient noise tomography (De Siena et al. 2018, GRL) shows significant lateral contrasts (Fig. 2b) even inside the caldera rim. The mention of the velocity log data is more compelling, but logging locations' scarcity means an incomplete sampling of the wider caldera.

The structural vs lithological question is interesting. One could be tempted to say that the relationship is exclusively structural given the correlation of the polarized patterns with the main directions of fractures (NW-SE and NE-SW) and the agreement with hypothesized extensional faults and transfer structures. However, this would be an obvious simplification. There is a clear correlation between the absence of polarization and the location where alteration is maximum: across the “metasediments” highlighted, e.g., by Siniscalchi et al. (2018) east of Solfatara. In recent research (*Di Martino et al. 2021, GJI, 226, Pages 1858–1872, not cited in the paper*), it has been proven that scattering (the primary mechanism through which polarization is lost, following recent results of Takagi et al. 2018) is strongly enhanced or dampened by differences in pore space and alteration in volcanic samples. All this considered, we believe that what is stated in the paper (L157-160) is sufficient to clarify the importance of lithological contrasts in changing polarization permanently.

Line 124 “when the central and eastern unpolarized reservoirs connect..”. This sentence sounds awkward. What’s an unpolarized reservoir? Please, re-phase.

We deleted the term unpolarized and rephrased as “when the central and eastern reservoirs (Fig. 1a) connect due to fluid injections and migrations (Figs. 1b, FIM), appearing as a single unpolarized anomaly” L156-157

Line 125 – “Fluids eventually outflow on metasediments”. Check English usage.

Rephrased as “Fluids eventually flow through metasediments ” L157

Lines 129-130 “The pre-seismic (Fig. 1c) and post-seismic (Fig. 1d) patterns show the progressive depolarization induced by fluids migrating from the injection location to...” How were pre-seismic patterns obtained?

The timing is expressed in the caption of Fig. 1c,d. As explained in the methods (*Data processing and estimates of horizontal polarization values*, referred to in the main text at L86), ambient noise was processed using the same stations before and after the seismic sequence, with the same procedure used for the year-long data.

Reviewer #2 (Remarks to the Author):

Dear Dr. De Siena and Dr. Petrosino,

Very interesting manuscript. Your article “Fluid migrations and volcanic earthquakes from depolarized ambient noise” studies how noise polarization can be used as a new tool to monitor pressurized fluids in volcanic systems, which may induce volcano-tectonic events and migration after seismic sequences. This technique can also be used for monthly and daily monitoring of fluid migration with minimum processing. The test case is focused on the Campi Flegrei caldera (South Italy), one of the most instrumented and long-term studied areas. Using low-frequency ambient noise (0.2-1 Hz), you found that the caldera-bounding extensional faults and a transfer structure crossing the deforming center and vents are consistently polarized during periods of low seismicity. When the seismic and geochemical activity increases, these structures, in particular the transfer feature, depolarize due to the migration of pressurized fluids which may trigger larger magnitude volcano-tectonic events (2019 and 2020 sequences).

Your paper presents a unique and novel approach to improve our understanding of volcanic eruptions forecasting and monitor volcanic unrest, as well as short-term monitoring of pressurized fluids. I think the method you propose here may also help the entire community to better understand volcano-earthquake triggering. Finally, I think this approach may be used in combination with other volcanic hazard forecasting tools, such as geochemical, remote sensing and other geophysical data.

I provide some comments and suggestions on the content, wording and presentation below, which I hope you will find helpful. For simplicity, I abbreviated the number of lines as e.g., L1 means Line #1:

We thank Simone Puel for the positive comments about our work and comments focused on improving readability and presentation. We are grateful for pointing out the relevance of the paper for the wider monitoring community and spending so much effort in checking text and figures. In the following, we address the remarks point by point, providing line numbers (L) corresponding to changes.

1. I am not sure to understand how exactly fluid migrations are monitored if the transfer structure depolarizes (L52). In Fig 1b-d I don't clearly see the fluid geometry, unless the fluids are only present where the value of R becomes low. Although I don't doubt that the polarization of both the transfer structure and in particular the extensional caldera-bounding faults are pretty consistent, from Fig. 1 I am not able to track how these fluids migrate to the east triggering the larger-magnitude earthquakes (circled number 1 and 2 in Fig. 1c and d). Maybe, further explanation may be needed.

The paper introduces the concept of “depolarized noise” – noise that showed polarization across given Earth structures and directions and loses said polarization once pressurized fluids interact with the structures. Noise is depolarized due to scattering from fluid injections and migrations, as proven by Saade et al. 2018, or across fluid-filled sediments (Takagi et al., 2018). Labels have been added to Figs. 1-3 to better mark this depolarization and help the reader understand the pathway of migrations - see Figs 1-3 and corresponding captions. Notice the letter *FIM* (fluid injection and migration) and *MF* (migrated fluids) in Figs. 1-3 and *ER* (expanding reservoir) in Fig. 2. For your specific question, in 2009 and 2017, the central reservoir (not polarized - *unpolarized*) and the eastern metasediments (also unpolarized) were separated by the transfer structure (*TS*). In 2018, they were connected after the *TS* depolarizes at Solfatara. Migrated fluids (*MF*) move to the metasediments and Monte Nuovo (M, western caldera) before the first earthquake and permanently move the eastern unpolarized anomaly further east after the sequence.

Hopefully, these marks help better understand the migration process, especially in Fig. 1. Explanations of all labels are given in the captions. We tried to keep these labels to a minimum as the guidelines of the Nature journal advice against imposing names on figure panels to avoid cluttering, in particular, they state (see <https://www.nature.com/nature/for-authors/formatting-guide>):

Where possible, text, including keys to symbols, should be provided in the legend rather than on the figure itself.

We leave the decision to reviewers and editor. It is easy for us to recreate figures with full names, like in Extended Data Fig. 1c,d.

2. L55-L57: it is not clear what the authors mean, since data from 1984 are plotted against data of 2009-2017. Therefore, I am not sure if there is a straight connection/correlation between the two datasets. I would reword the sentence as: “The high-attenuation signature of ... seems to coincide with the area of low polarization, still suggesting actual presence of pressurized fluids within the same region (Fig. 2a)”. If, instead, the authors suggest a sort of correlations between the actual unpolarized anomaly and the high attenuation of three decades ago, my questions would be: what may be the physical explanation or evidence to justify a temporal correlation between them?

We added a more precise explanation of why the two periods are connected and the 1984 seismicity is presented at L52-58. In summary, the answer is contained in the three cited papers (De Siena et al. 2017, SR and De Siena et al. 2017, GRL; De Siena et al. 2018, GRL). They use earthquake locations and coda attenuation tomography and find that the area of maximum attenuation (Fig. 2a) corresponds to the location of repeated vertically aligned seismic clusters. The timing of these clusters corresponds to that of injections modelled by Chiodini et al. 2017 Nature: Communication with THOUGH simulations from a deep source of magmatic fluids. In De Siena et al. (2018), it is proven that the area is comprised in the reservoir of fluids detected by Vanorio et al. 2005 (JGR) in 1984. This reservoir had expanded laterally and aseismically in 2011-2013. Injections of fluids are known to change seismic scattering dramatically, and the plot serves to demonstrate that the location of the 1984 injection remains unpolarized in 2009 and 2017.

3. L65-66: the high-velocity/stress barrier doesn't exactly coincide with the transfer structure of Fig. 1a. In Fig. 2b we can see many low R areas in the studied region. It can be a smaller scale as shown in Fig. 2b with perpendicular polarization to the faults. I would suggest to say that the transfer area is located in the area of high velocities and stresses. Otherwise, how would they explain the low depolarization calculations (yellow dots in Fig. 2b) within the high-velocity structure?

We do not expect the two to coincide, given the wavelengths in play (of the order of ~1 km) and especially the lateral resolution of the ambient noise tomography (see De Siena et al. 2018). Yet, we focus on the WE pattern of low-high-low velocity going from east of Monte Nuovo to the metasediments and caldera-bounding faults. The transfer structure is a smaller-scale feature in the high-velocity zone. Also, there is a clear spatial relation with the SN barrier derived from InSAR data. To find a compromise, we wrote at L95-97:

“The transfer structure crosses the high-velocity eastern sector of the Solfatara crater¹³ and borders the SN anomaly that accumulates the highest lateral stress and produces seismicity during unrest^{32,40} (Fig. 2b, white ellipsoid).“

4. In Fig. 1a it is not clear whether the analyzed data are collected in the period between 2009 and 2017 or only 2009 and 2017. In L99 it says in between, while in the caption of Fig. 1 and 2 it seems the data are two distinct datasets, only from 2009 and 2017, respectively.

The mistake is at L99 in the previous version. The data are from the two distinct years, which have shown the lowest seismic release across the decade. We noted other places where this was ambiguous and corrected it accordingly; thanks for noticing. We clarified it in the new introduction at L86.

5. L119-120: I think the authors may remove the second part of the sentence "..., adding persistent low-frequency noise and long-period events". The reason is that it makes confusion to the reader with the next sentence. If the migration of pressurized fluid tends to depolarize the structure, it indicates that their signals can cause high-frequency noise, not long-periods events. Otherwise, I would reword the sentence as: "..., adding high-frequency components to a background low-frequency noise and long-period events".

There is probably confusion here about the processes we are trying to describe. We do not want to state that the fluids are adding high-frequency noise on a low-frequency background. The 0.2-1 Hz is the typical band for long-period seismicity and hydrothermal noise. While there has been discussion about this (Bean et al. 2014, Nature Geosciences), fluids are still considered the primary cause of LPs. We believe that mentioning the high-frequency noise would cause a trade-off with the results obtained in the high-frequency band (1-5 Hz). To clarify the concept, we rephrased:

"These geophysical responses and maps are linked to the sub-caprock migration of over-saturated pressurized fluids¹⁴ of hydrothermal^{17,18,23} or magmatic^{19,21,46} origin, which produce persistent low-frequency noise and long-period events at the caldera⁴⁴." L149-151

6. L124: I would change "apparent" with "seems to occur", since apparent can have two meanings, one as a synonym of "clear/obvious" and the other as "maybe true, but not necessarily". I am not sure which version of the term the authors mean.

We agree that apparent is ambiguous in this context. We changed it to :

"... this behaviour manifests itself.." L155.

7. L125: I am a bit surprised about the role of the metasediments. While I may understand how sediments may reduce ambient noise, I am not sure whether these sediments are present only in the last decade (Fig. 1) and not present before (I would think they were also there three decades before). One thing it would be very interesting to look at is the distribution of lithotypes in the geologic map of the area and compare it with the polarization results, in order to confirm whether or not there is a spatial correlation between structures and lithology (metasediments). Only in this case, this cause-effect would be consistent.

The analysis is restricted to the last decade as there were no noise data available before. While the sediments were likely there in the past, their level of alteration was probably different before 2009. This inference is related to increased flow and geochemical interactions, which have been reported by geochemical studies and are testified by the opening of the Pisciarelli vent in 2010. This point has been brought forward in a different form by reviewer 1, which goes the opposite direction and asks if we could not interpret the results exclusively in lithological terms. We report below the answer:

The structural vs lithological question is interesting. One could be tempted to say that the relationship is exclusively structural given the correlation of the polarized patterns with the main directions of fractures (NW-SE and NE-SW) and the agreement with hypothesized extensional faults and transfer structures. However, this would be an obvious simplification. There is a clear correlation between the absence of polarization and the location where alteration is maximum: across the "metasediments" highlighted, e.g., by Siniscalchi et al. (2018) east of Solfatara. In recent research (Di Martino et al. 2021, GJI, 226, Pages 1858–1872, not cited in the paper), it has been proven that scattering (the primary mechanism through which polarization is lost, following recent results of Takagi et al. 2018) is strongly enhanced or dampened by differences in pore space and alteration in volcanic samples. All this considered, we believe that what is stated in the paper (L157-160) is sufficient to clarify the importance of lithological contrasts in changing polarization permanently.

The research of Takagi et al. (2018) offers support to a relation between lithology and scattering in sediments that depolarizes noise. We believe we have enough ground to state a relationship between polarization and lithological contrasts, as at L157-160.

8. L140-146: I would re-organize this paragraph as follows: “Gas-bearing fluids over-pressurized the eastern caldera between 2011 and 2013 due to concurrent lateral expansion of the source region and saturation of the reservoirs. This caused the highest horizontal stress in the area east of the Solfatara feeder (white diamond, Fig. 2b,c). The depolarization of the transfer structure that started in 2018 (Fig. 1b) marks the process that led to the highest seismic release in thirty-six years at the caldera. Here, fluid injections from depth coupled with progressive permeability increases from heavy rains started the seismic sequence in December 2019”. In this way, it seems to appear a temporal cause-effect.

We thank you for improving the paragraph and accept your suggestion. L174-180

9. L152: I would put: “... which horizontal noise polarization can monitor. For example, the mechanical weakening ...”. Otherwise, the reader may wonder what the correlation between Campi Flegrei and the Tohoku-oki earthquake is.

As before, thanks for the suggestion. L185

10. L163-165: I would rewrite this sentence because it misses some subjects. I am not sure what are the subjects of “mostly likely to erupt, form new hydrothermal vents, and nucleate earthquakes”. Are the velocity reductions? Fluids? Area? Moreover, I am not a fan of the word “symptomatic”. You may use another word like “denote/indicate”.

The confusion is due to our attempt to describe two different processes, i.e., that velocity reductions mark areas of pressurized fluids AND that, at Campi Flegrei, these areas pose the highest hazard. We rephrased as:

“These reductions indicate the area bearing the highest concentration of pressurized fluids^{10,11}, which is also the part of Campi Flegrei most likely to form new hydrothermal vents and nucleate earthquakes^{51,52}.” L196-198.

11. L182: “sealed the migration...”.

Ok, L215

12. L195: I would change “Once informed by ...” with: “Through thermo-...”.

Considering the sentence written just after this one, we decided to rephrase the period as: “Knowing the delay between deep fluid injections and activation of faults in the eastern caldera allows us to investigate the real-time potential of depolarized noise. This delay was obtained by recent thermo-hydro-mechanical modelling⁴⁸.” L228-230.

13. Extended Data Figure 1: Just a general comment/suggestion. Although I know that the number of stations is limited and concentrated in a small and narrow region, the squared interpolation from each station can cause not realistic artefacts, in particular in regions where there are no data (for example, in the north-east, west, south, and south-east of the studied region). Despite the fact that in the high-station region the

interpolation may be close to the pointwise measurement, I would suggest removing the interpolation in areas where there are no data.

Reviewer 3, Thomas Lecocq, also made this suggestion. We created a new Extended Data figure 7, which shows the same maps in Figures 1 and 3 plotted with a new interpolation. We used a maximum distance for the squared interpolation equal to the wavelength, given by the product of the average shear velocity derived from ambient noise (1.2 km/s) multiplied by the dominant frequency in our band (0.7 Hz): $\lambda=1.7$ km. We only interpret features appearing in all maps. It is central for us to have values of resultant length for all points in space, as these are used to populate the wave equation modelling. As written at L91-92:

“Here and in the following, we only interpret features that are independent of mapping interpolation (Extended Data Fig. 7).”

14. L272 (caption): I think there is a typo. It is not “b)” but “c-d)”.

Yes, thanks, L313.

15. L275-277: Again, it is unclear if the authors mean between 2009 and 2017, or the two distinct years.

As stated before, it is 2009 and 2017, so in this case the concept is correct.

16. Extended Data Figure 4: it misses the colorbar. Personally, I think that this plot is very interesting since it corroborates the hypothesis to use noise depolarization for real-time monitoring.

Thanks for noticing, colorbar added. Yes, we are quite confident we can obtain stable measurements over one hour. However, there remain obvious questions. Does the polarized structure that divides the caldera between west and east over a single hour (and depends on just one station) mark active processes? Or is it just due to unstable processing over a short period, thus disappearing over six months? Previous work of the lead author proves that the technique is exceptionally stable over months and can be applied up to one day. At present, we believe we have no sufficient ground to prove stability over one hour. Yet we state with more confidence our ability to retrieve the extensional faults with just 1 hour of noise at L85.

17. L309: I think there is a typo. It should be “January-March 2020” not 2019 (at least from the plots below).

Yes, thanks, L351.

18. L313-314: I don’t understand the following sentence: “..., where the intervals have been interpreted using the results of thermo-hydro-mechanical modelling”. I would suggest removing this sentence or provide further information regarding the choice of the time intervals.

This is now better explained in the main text (L228-230) thus we removed the sentence.

19. L339: Again, it is unclear if the authors mean between 2009 and 2017, or the two distinct years.

Here we were wrong again, stating it was 2009-2017 instead of 2009 and 2017.

20. L340-342: I don't quite understand these two sentences. The decrease of what? R ? Moreover, what is the lowest? R ? Because in Fig. 7a I can see in the brackets that the lowest R seems to occur for the computations where the sources are located at the coastlines, not in the far-field (0.9942 and 0.9983, respectively).

Any finite-difference simulation of the wave-equation is affected by instabilities, which are due to the finiteness of the differences and the boundary conditions. For a simulation without these numerical instabilities, we would expect perfect polarization at all stations ($R=1$) in both homogeneous cases (for far-field and coastline sources). We observe minimal depolarization (shown in the re-labelled Extended Data Fig. 8) for the homogeneous case: this gives us the threshold to interpret depolarization once we include heterogeneity, and test if simple structural heterogeneity is enough to reproduce measurements.

We rewrote at L390-398: "The decrease in the homogeneous cases (Extended Data Fig. 8a) is due to numerical instability, the finiteness of the differences, and boundary conditions only⁵¹. Depolarization (reductions of R) in the homogeneous case and at these frequencies are minimal (below 1%) at all stations for both source configurations (Extended Data Fig. 8a). ... This gives us a threshold to interpret if the sole existence of velocity contrasts can reduce R at the levels observed in the data."

21. L343: I figured out that "L.D.S." and "S.P." refer to the name of the authors, but only after reading the "Author contributions". Personally, I would remove the names and make more an informal description of the method, something like: "The polarization parameters are retrieved with a blind test, where synthetic seismograms are processed without inputs on the original polarization".

We accepted the suggestion, thanks, L393.

22. L351: I wonder why the authors choose to increase 50% of shear modulus only in the regions of $R>0.31$. Where does the choice of $R=0.31$ come from? How does the result change if we choose $R>0.4$? What may be a physical justification of 0.31?

We set the 0.31 threshold experimentally, as this is the average polarization value over the entire dataset in 2009 and 2017. This is now made explicit at L402-403. The 50% is obtained by De Siena et al. 2018, as high polarization corresponds on average to high velocities.

23. Extended Data Figure 7: Since the residuals are pretty high and very similar between the homogeneous and heterogeneous case, I wonder how much panel 7b (heterogeneous case) adds to the paper. Visually comparing the two tests, I cannot see a very clear difference between them to justify the need of making the model more complex. The residuals do not improve much. Personally, I would leave the panels 7b out and comment in the caption that reasonably changes in shear modulus locally do not significantly improve the results/conclusions with respect to the homogeneous case.

This is exactly the point. The inclusion of velocity heterogeneity alone is insufficient to decrease R at the level we observe in the data unless near-station extended sources are active (station ACL2). This combination creates trapped and scattered waves that reduce R to levels comparable to data. Also, the existence of heterogeneity does not improve residuals enough to state that azimuths are clearly dependent on heterogeneity in the absence of anisotropy, as they do not reduce residuals enough.

We decluttered Extended Data Figure 8 and now write at L413-415:

“However, at least for the simulated isotropic case for this frequency band, the sole existence of high-velocity heterogeneity as observed at Campi Flegrei has only minor effects on azimuths: these are primarily controlled by the location of noise sources.”

24. L380-383: I am not completely sure about that at low-frequencies (0.2-1 Hz) we can see a clear decrease or increase of R . From the left panel (second plot from the top) the values of R do not clearly deviate. Although there are local decreases and increases (e.g., after the small swarm at Pozzuoli), there is not a very consistent and evident deviation as seen in the panel on the right (1-5 Hz). In this case, I agree that there is a permanent decrease of R after the Monte Nuovo swarm.

Just looking at the figure the change is not as apparent as at high frequencies, but our statistical test confirms it between Jun 2012 and Jan2013. See L435-439.

“Between 0.2 Hz and 1 Hz (Extended Data Fig. 9, left) the hypothesis of the unequal mean is confirmed only considering data between June 2012 and January 2013. The cause is a small swarm in April 2012³⁰ that decreases R temporarily. After this swarm we observe a progressive low-frequency increase of R , indicative of pressurization of the deeper systems.”

25. Extended Data Figure 8: There are two vertical lines in the left panel (0.2-1 Hz). What do they refer to? They are not defined either in the caption or in the text.

This is now ED Fig. 9. The lines are a typo that has been removed. Following the suggestions of reviewer 3, we have also modified the figure to better highlight the variations of R .

26. Extended Data Figure 9: I would put the labels (a and b) to the plot. In L390 I am not sure what “... and two and ...” refers to? In addition, the last sentence: “The variations indicate that the parameter is sensitive to the fluids released by the earthquakes” may make the reader wondering why and at what degree the parameter change is sensitive to the fluids, in particular those released by the seismic sequences. Which type of fluids do the author refer? At least, it is not completely clear to me. Further explanation may be needed.

This is now ED Fig. 10. We have added the labels (a, b) and re-arranged the panels to show first the p-values. At L390 “two and” was a typo, we have corrected it.

We apologize for the unclear caption, which we have rewritten to better explain the results of the statistical t-test. The test allows to detect if there is a statistically significant variation of the tested parameter (in this case, the resultant length R), but of course it does not tell us the cause of the variation. The interpretation of the changes in R is discussed in the main text at L198-206, in terms of fluid migration and consequent stress build-up and release through the seismic sequences.

Hope these suggestions will help to improve the manuscript. I Look forward to seeing this published.

Best wishes,

Simone Puel

Reviewer #3 (Remarks to the Author):

Dear Authors,

I've read your paper "Fluid migrations and volcanic earthquakes from depolarized ambient noise" with great interest. I find your method beautiful in its simplicity, allowing to track migrations in one of the most volcano-hazardous regions in the world.

The manuscript is well organized and written, and easy to follow.

We thank Thomas Lecocq for the appreciating our works and the comments focused on improving readability and presentation. We are grateful for pointing out the simplicity of the methodology, especially the fact that it requires little assumptions and processing. We are also thankful for pointing out that the manuscript is well written and organized: you will notice we changed to a more conventional Introduction/Results and Discussion format tackling Reviewer 1 comments but tried to keep the flow of the paper untouched. In the following, we address the concerns point by point, providing line numbers (L) corresponding to changes.

My main remark concerns the figures, which are difficult to read at first sight. I think the authors should define a maximum distance from a station their "squared interpolation" is no longer trustable, and mask those areas, this would give highlight where the map is really supported by data and where not.

Reviewer 2, Simone Puel, also made this suggestion. We created a new Extended Data figure 7, which shows the same maps in Figures 1 and 3 plotted with a new interpolation. We used a maximum distance for the squared interpolation equal to the wavelength, given by the product of the average shear velocity derived from ambient noise (1.2 km/s) multiplied by the dominant frequency in our band (0.7 Hz): $\lambda=1.7$ km. We only interpret features appearing in all maps. It is central for us to have values of resultant length for all points in space, as these are used to populate the wave equation modelling. As written at L91-92:

"Here and in the following, we only interpret features that are independent of mapping interpolation (Extended Data Fig. 7)."

I think the article lacks a small discussion on how this polarization analysis could be useful (or not) for other locations, where caprock is (or not) present?

We believe that the existence of the caprock and the high lateral stress recorded at the caldera are central for the stability of our maps. Without stressed high-polarization structures we cannot detect depolarization! Results will be more unstable when magma creates new pathways in the shallow crust, thus complicating the network underlying the volcano, or for different lithologies. Topography (e.g., for stratovolcanoes) could also diminish the reliability of connecting faults with polarized noise. This said, the technique seems ideal for calderas with relatively long time of quiescence, like e.g. Redoubt.

We added the following sentence to the discussion (L253-258):

"Both the caprock and associated high lateral stress at the caldera seem crucial for monitoring volcanoes with noise depolarization. Without persistent high polarization across structures that bear most of this stress, it is unlikely to discriminate depolarization from processing uncertainties. While this could be an important limitation at volcanoes that release stress frequently, like Etna, and that present different lithological contrasts, the technique seems ideally suited to image and monitor volcanoes with long periods of repose."

Small remarks:

- Figure 2c: what is the source of the Resistivity data?

The resistivity data were published by Siniscalchi et al. (2018, JGR) and provided to us directly from the first author (see Acknowledgement). The data are uploaded in the public dataset along with the codes necessary to reproduce the figure – see Materials and Correspondence – and mentioned in the caption.

- Figure 2c: what are the uncertainties on the event locations?

The earthquakes have uncertainties lower than 0.2 km for both events as calculated by the INGV. Such low uncertainties are possible as the earthquakes happen in the area best covered by seismic stations. We added reference 38 (Bellucci Sessa et al. 2020, ADGEO) when the earthquakes are mentioned in the captions, where locations are obtained.

- L200: "depolarization": does it mean "absence of significant polarization"? from "de-" it feels like it was polarized before, then got DE-polarized ?

Exactly! The paper brings upfront the concept of “depolarized noise” – noise that showed polarization across highly-stressed Earth structures and directions and loses said polarization once pressurized fluids interact with the structures. This concept is now better explained at L30-37 after comments from reviewer 1. The word was previously used only in the context of potential electromagnetic fields.

- L216: not sure "compressed" is the best word here, do you mean something like "sandwiched"? or "located" ?

Changed to “sandwiched” (L250), thanks.

- L219: "instantaneous": from the method and the results shown, it seems indeed possible to have real time information

Yes, we are quite confident we can obtain stable measurements even over one hour. For example, in Extended Data Fig. 3 we compared data obtained over 6 months and 1 hour in 2018, obtaining equivalent results for the extensional faults. Previous work of the lead author proves that the technique is exceptionally stable over years and months and can be applied to time series up to one day long. As we know well the caldera, where there is now real-time monitoring of volcanic gas and deformation, we have alternative data to interpret the map at that precise time and day.

However, obvious questions remain. Does the polarized structure that divides the caldera between west and east over a single hour (and depends on just one station, Extended Data Fig. 3) mark active processes? Or is it just due to unstable processing over a short period, thus disappearing over six months? At present, we believe we have no sufficient ground to prove stability of depolarization over one hour. Yet, we state with more confidence our ability to retrieve the extensional faults with just 1 hour of noise at L86.

- L224: "minimal processing": with the comment for L219: I think it'd be worth quantifying the time/processing needed indeed.

At present, the primary problem is a workflow lacking optimization. For 1 hr of noise registered at the maximum number of stations used (37) the polarization parameters are computed in a few minutes on a standard laptop. However, most of the time is lost in preparing “clean” SAC input data from the monitoring network of the observatory, visual quality checking, and conversion. Observatories often change instruments and sometimes locations of stations, so it was necessary to check a lot of metadata in different formats to be sure that, e.g., all stations used here were broadband. You can have a look at the Matlab codes employed in the repository <http://doi.org/10.17605/OSF.IO/KQTBP> (see the Data Processing Folder).

- L232: "GMT" -> "UTC"

Corrected throughout the text, thanks.

- L252: 117h vs 171h: the authors love to play with similar stuff (Md3.1 & Md3.3 too) :-)

But the numbers are correct (117 hr for 2019 and 171 hr for 2020), even if it is weird that the last two numbers are switched!

- L254-259: It's unclear to me if the authors actually use the vertical component in the covariance matrix method, and if yes, does the incidence angle of the dominant polarization provide any interesting information?

The covariance matrix is calculated using the three components of motion. The matrix is diagonalized and its eigenvalues and eigenvectors define the axis length and orientation of the polarization vector. Then, we used the azimuth of the polarization vector (the angle between the geographic north and the projection of the main eigenvector on the horizontal plane) associated with an incidence angle $>45^\circ$, which corresponds to a high horizontal polarization degree. See Cusano et al. 2020, JVGR, for a recent publication.

Regarding the incidence angle, we looked at its spatial distribution, which appears uniform through space, confirming the possibility of modelling noise polarization variations primarily on the horizontal plane (Extended Data Fig. 8).

- L307: "circled number 1" doesn't refer to the Figure where it's drawn

We believe it does, in the section we discuss Extended Data Fig. 5, and the earthquake appears in December-January of this Figure.

- L343: weird location to put the authors' contributions

They have been removed, also following the remark of Reviewer 2 that this is simply a blind test. See L393.

- Extended figure 8: what is the source of the events' location ? The key changes should be highlighted on the top two panels (circle where the changes occur, draw trend lines before-after the MN Swarm)

The 2012 seismic sequence has been studied in D' Auria et al., 2015, SR and Bellucci et al., 2021, ADGEO. Locations are provided by both the authors. We added the proper references.

We have modified Extended Data Fig, 8 (now Extended Data Fig. 9), drawing trend lines and evidencing the variations of the resultant length. We discussed the reference for the events' locations before (Bellucci et al. 2020).

"Data Analysis software" can be asked to LDS : why not putting it on the author's github?

The sections Data Availability and Code Availability are now included at L455-471. All the software used to recreate the figures in the paper was included in the Open Science Framework (OSF), link <http://doi.org/10.17605/OSF.IO/KQTBP>. Now we also included the data analysis software, which has been developed by the lead author. The script for the polarization analysis is also available in a Zenodo repository at <http://doi.org/10.5281/zenodo.5131531>. The seismic modelling tool (Extended Data Fig. 8) has been also uploaded by the second author on his GitHub page at <https://github.com/LucaDeSiena/SH-Ambient-Noise-NatComm>. See L455-469.

Please cite all the software used to read/process/plot data, including the libraries & GIS software.

We used primarily Matlab[®] for data processing and visualization, and Golden Software Grapher[™] to produce Extended Data Figs. 9-10. The shape files are uploaded at osf.io/kqtbp and were published by references [24-26]. Code Availability is now stated at L462-471.

Looking forward to seeing this article published,

Best regards,

Thomas Lecocq

REVIEWERS' COMMENTS

Reviewer #1 (Remarks to the Author):

Thank you very much for your efforts in improving the manuscript from the last submission. The authors addressed most of the comments from the previous review. The writing has greatly improved, the methodology is well described, and overall, the paper is well organized and much easier to comprehend. I feel that the paper is ready for publication.

Reviewer #2 (Remarks to the Author):

Dear Dr. De Siena and Dr. Petrosino,

Thank you so much for the revision of your manuscript. Now, I believe that the entire paper is better re-designed and written than the previous version. Very good job! I also love the sentence L253-258 talking about the limitations of the technique.

All my comments and suggestions have been addressed by the authors, and I look forward this manuscript to be published. In the following, I address point by point, providing my final small suggestions to further improve the written.

- 1) I think that the labels help the reader to visualize the different structures and the migration of fluids to the western caldera.
- 2) My second point has been answered by the authors.
- 3) Thanks for the explanation. I think now it is clearer.
- 4) Now it is clear.
- 5) Yes, I agree with the correction, now it is much clearer. Thanks.
- 6) I would change "manifest itself" with "This behavior may be visible".
- 7) The explanation provided by the authors is sufficient to state the importance of the relationship between lithology and polarization. Maybe in L157 or L158, I would add "altered metasediments" to insist on the importance of alteration according to the authors' explanation: "There is a clear correlation between the absence of polarization and the location where alteration is maximum: across the "metasediments" highlighted, e.g., by Siniscalchi et al. (2018) east of Solfatara".
- 10) The new sentence is much clearer than before, thanks.
- 12) Now it is clear.
- 13) I think the Extended Data Figure 7 is necessary. I would rephrase L91-92 in something like: "The results and the transfer structure discussed in this paper are independent of mapping interpolation (Extended Data Figure 7)", otherwise the sentence before can be misunderstood, requiring the reader to see the extended figure right away.
- 18) Now it is much clearer, thanks for removing the sentence.
- 20) Now it is clearer, thanks for the explanation.
- 22) Thanks for explicitly write the explanation; I think now it is clear to the reader the choice of $R = 0.31$.
- 23) I agree with removing the values in brackets in Extended Data Figure 8. Both the figure and the added sentence help the reader to understand the point of the authors.
- 24) I think the green lines in Extended Data Figure 9 now help the reader to note the decrease in R after the Monte Nuovo swarm, also for 0.2-1 Hz frequency.
- 25) OK, thanks for the modification, now it is much clearer.
- 26) Thanks for the further explanation in the caption. Now, I think the point of the authors is clearer.

Finally, my last suggestion is to remove "and" in L47. I would rewrite L47 as: "Fieldwork, numerical modeling, and ...".

Hope these final suggestions will help to improve the manuscript. I Look forward to seeing this published.
Best wishes,
Simone Puel

We thank all reviewers for the time and effort spent improving our work: it is greatly appreciated. We are particularly grateful to Simone Puel for the additional comments aimed at strengthening presentation. In the following we give an account of all the changes done to address his points and format the manuscript for publication.

Formatting

As requested by the Author Checklist, we renamed all *Extended Data Figures* to *Supplementary Figures*. These Figure and the Supplementary Table are included in a new file named "Supplementary Information".

We redrew all figures of the manuscript to satisfy the formatting requirements of Nature: Communications. The primary change is that now all figures have a legend, so captions have been reduced. This also account for the need of better explanations of where exactly the transfer structure or extensional faults are.

Response to Simone Puel2:

We thank you for the additional comments aimed at strengthening the presentation. Here we answer to the standing points:

1) I think that the labels help the reader to visualize the different structures and the migration of fluids to the western caldera.

1) Labels and legends have been added in all figures, following Nature: Communications guidelines.

6) I would change "manifest itself" with "This behavior may be visible".

6) We believe the sentence can be further clarified, so we wrote: **This behaviour could be the cause of the depolarization of the transfer structure at Solfatara in 2018, when fluid injections and migrations acted as extended sources and connected the central and eastern reservoirs (Figs. 1b).**

7) The explanation provided by the authors is sufficient to state the importance of the relationship between lithology and polarization. Maybe in L157 or L158, I would add "altered metasediments" to insist on the importance of alteration according to the authors' explanation: "There is a clear correlation between the absence of polarization and the location where alteration is maximum: across the "metasediments" highlighted, e.g., by Siniscalchi et al. (2018) east of Solfatara".

7) We followed the suggestion.

13) I think the Extended Data Figure 7 is necessary. I would rephrase L91-92 in something like: "The results and the transfer structure discussed in this paper are independent of mapping interpolation (Extended Data Figure 7)", otherwise the sentence before can be misunderstood, requiring the reader to see the extended figure right away.

13) We used the suggested sentence.

Finally, my last suggestion is to remove “and” in L47. I would rewrite L47 as: “Fieldwork, numerical modeling, and ...”.

We removed the word “and”